# Coral Restoration Effectiveness: Multiregional Snapshots of the Long-Term Responses of Coral Assemblages to Restoration

**Margaux Y. Hein [1,2,*], Roger Beeden [3], Alastair Birtles [1], Naomi M. Gardiner [1], Thomas Le Berre [4], Jessica Levy [5], Nadine Marshall [6], Chad M. Scott [7], Lisa Terry [8] and Bette L. Willis [1,2]**

1   College of Science and Engineering, James Cook University, Townsville, QLD 4811, Australia; alastair.birtles@jcu.edu.au (A.B.); naomi.gardiner@jcu.edu.au (N.M.G.); bette.willis@jcu.edu.au (B.L.W.)
2   Australian Research Council (ARC) Centre of Excellence for Coral Reef Studies, Townsville, QLD 4811, Australia
3   Great Barrier Reef Marine Park Authority, Townsville, QLD 4811, Australia; roger.beeden@gbrmpa.gov.au
4   Reefscapers Pvt Ltd., Seamarc, Marine Discovery Centre, Landaa Giraavaru 20215, Maldives; thomas@reefscapers.com
5   Coral Restoration Foundation, Key Largo, FL 03037, USA; jessica@coralrestoration.org
6   CSIRO Land and Water, ATSIP Building#145, James Cook University, Townsville, QLD 4811, Australia; nadine.marshall00@gmail.com
7   Conservation Diver, Evergreen, CO 80439, USA; info@conservationdiver.com
8   The Nature Conservancy Caribbean Program, 3052 Estate Little Princess, Christiansted, VI 00820, USA; lisa.terry@tnc.org
*   Correspondence: margaux.hein@my.jcu.edu.au

**Abstract:** Coral restoration is rapidly becoming a mainstream strategic reef management response to address dramatic declines in coral cover worldwide. Restoration success can be defined as enhanced reef functions leading to improved ecosystem services, with multiple benefits at socio-ecological scales. However, there is often a mismatch between the objectives of coral restoration programs and the metrics used to assess their effectiveness. In particular, the scales of ecological benefits currently assessed are typically limited in both time and space, often being limited to short-term monitoring of the growth and survival of transplanted corals. In this paper, we explore reef-scale responses of coral assemblages to restoration practices applied in four well-established coral restoration programs. We found that hard coral cover and structural complexity were consistently greater at restored compared to unrestored (degraded) sites. However, patterns in coral diversity, coral recruitment, and coral health among restored, unrestored, and reference sites varied across locations, highlighting differences in methodologies among restoration programs. Altogether, differences in program objectives, methodologies, and the state of nearby coral communities were key drivers of variability in the responses of coral assemblages to restoration. The framework presented here provides guidance to improve qualitative and quantitative assessments of coral restoration efforts and can be applied to further understanding of the role of restoration within resilience-based reef management.

**Keywords:** coral assemblages; coral restoration; effectiveness; monitoring

## 1. Introduction

The number of coral restoration programs is burgeoning in most reef regions in response to worldwide declines in coral cover in recent years [1–3]. Common objectives of these programs are

to assist the recovery of reefs, protect endangered coral species, promote sustainable alternative livelihoods, and enhance conservation stewardship [4], but there is a general mismatch between the stated objectives of these programs and indicators used to assess their effectiveness. In general, most assessments of coral restoration effectiveness are largely focused on the number of ramets created, growth, and survival post-transplantation [4]. A recent review of coral restoration efforts globally revealed a lack of appropriate and standardized monitoring of outcomes, with too short timeframes (median monitoring time of 12 months) to assess the potential of using restoration as a tool for resilience-based management [5]. This lack of long-term comprehensive assessments of coral restoration effectiveness is widely criticized [6,7] and hinders the uptake of coral restoration within multi-scale resilience-based management frameworks [8]. In addition, many studies are focused on site- or region-specific restoration programs [9,10], which has made comparative studies difficult and limited the development of broad best-practice recommendations.

Improved resilience of degraded reefs is the ultimate objective of many coral restoration programs. Not only has "managing for reef resilience" become a major focus of reef management [8,11], but "re-establishing a self-sustaining, functioning coral reef ecosystem after a disturbance" is also the most commonly stated objective for coral restoration [4]. However, measuring the resilience of an ecosystem is a difficult exercise that requires a range of metrics accounting for aspects of both recovery and resistance over time [12]. Reef attributes like hard coral cover, species diversity, and structural complexity are directly related to reef resilience [8,11,13] and may be enhanced by restoration programs. Percent hard coral cover is the most widely used metric to document reef recovery (e.g., [14]), although its use in isolation has limited value [11,15,16]. At restoration sites, increased hard coral cover may prevent phase shifts to algal-dominated systems [15], enhance the recruitment of juvenile corals to damaged areas [17], as well as regenerate the structural complexity of a degraded reef [18]. Structural complexity may also be increased directly by artificial structures used as surfaces for coral transplants. High structural complexity of reef systems has been shown to decrease the sensitivity of local coral assemblages to extreme weather events [19], and also improve reef recovery post-bleaching [20]. Increased coral diversity on restored reefs may lead to increased biodiversity of associated vertebrates and invertebrates, and hence increased functional diversity present within the reef community [21]. Increased functional diversity may increase the resistance of the reef community by expanding the range of its potential responses to disturbances [8]. Assessing the potential for reef restoration to improve reef resilience thus necessitates looking at processes occurring at the scale of the benthic community rather than solely at the scale of coral fragments transplanted to a degraded reef.

A set of six ecological indicators that could be used to characterize the resilience of a reef community, based on an evaluation of indicators used in terrestrial restoration (e.g., [22]) and reef resilience studies (e.g., [13]), were developed by [4]. These are: (1) Coral diversity, (2) herbivore biomass and diversity, (3) benthic cover, (4) recruitment, (5) coral health, and (6) structural complexity. Although subsets of these indicators have been used to characterize the resilience of reef communities [11,13], a collective set of these indicators has not yet been applied to assessing the outcomes of a coral restoration program. While the capacity of a coral restoration program to affect one or more of these indicators positively is likely to be constrained by factors, such as the degradation state of the reef area to be restored or the types of strategies used to restore the coral community, in combination, they provide a holistic assessment of restoration effectiveness.

The objectives and methodologies of coral restoration programs typically differ among reef regions. In the Caribbean, most programs aim to restore two critically endangered species of Acropora (e.g., [23]), while programs in the Indo-Pacific are more focused on restoring reef structure and resilience [4]. Many programs depend on the capacity of corals to reproduce asexually and use either fragments from donor colonies or fragments of opportunity. While each methodology has its strengths and limitations in differing contexts, there is a critical need to further our understanding of how these different methodologies impact the resilience of restored reef areas in the long term to better inform reef managers.

The aim of this paper was to capture a snapshot of the responses of coral assemblages to long-term restoration practices at four locations with well-established coral restoration programs that differ in objectives, methodologies, and socio-cultural settings. At each reef location, five coral-based indicators of reef resilience were characterized: Coral cover, structural complexity, coral diversity, coral juveniles, and coral health. These five indicators of restoration effectiveness were then qualitatively compared among the four restoration programs to gain insights into how different restoration designs influence the response of coral assemblages to coral restoration. The response of fish assemblages to these coral restoration efforts were examined independently.

## 2. Materials and Methods

### 2.1. Study Sites

The four restoration programs selected had been in operation for 8 to 12 years, enabling assessments of the long-term effectiveness of differing restoration approaches. The programs selected represented four reef regions: (1) New Heaven Reef Conservation Program (NHRCP) on the island of Koh Tao, Thailand; (2) Reefscapers program on the island of Landaa Giraavaru, Maldives; (3) Coral Restoration Foundation in Key Largo, Florida Keys, USA; and (4) The Nature Conservancy on the island of St Croix, US Virgin Islands (Figure 1). Each location has a unique history of reef-associated disturbances; therefore, objectives for coral restoration varied from growing and restoring endangered species of corals (Florida Keys and St Croix), to restoring coral abundance and diversity at sites degraded by tourism pressures and bleaching events (Koh Tao and Landaa Giraavaru). Programs also differed in the set of coral restoration techniques used (see boxes 1–4; summarized in Figures 1 and 2), which provided an opportunity to qualitatively compare the relative effectiveness of different methodologies across the five indicators of reef resilience.

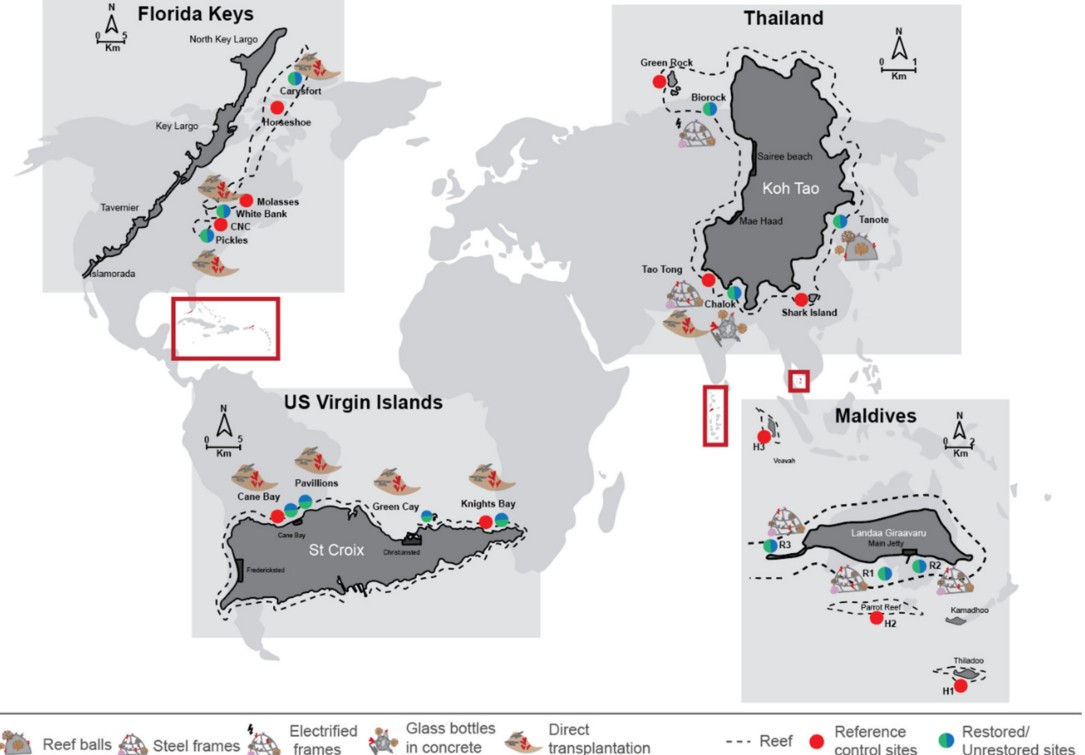

**Figure 1.** Map showing the locations of the four coral restoration programs surveyed and an overview of the restoration strategies used in each program (see key at bottom of figure to interpret diagrams that represent techniques present at each site). Half green and half blue circles indicate adjacent restored and unrestored sites; red circles indicate control reference sites.

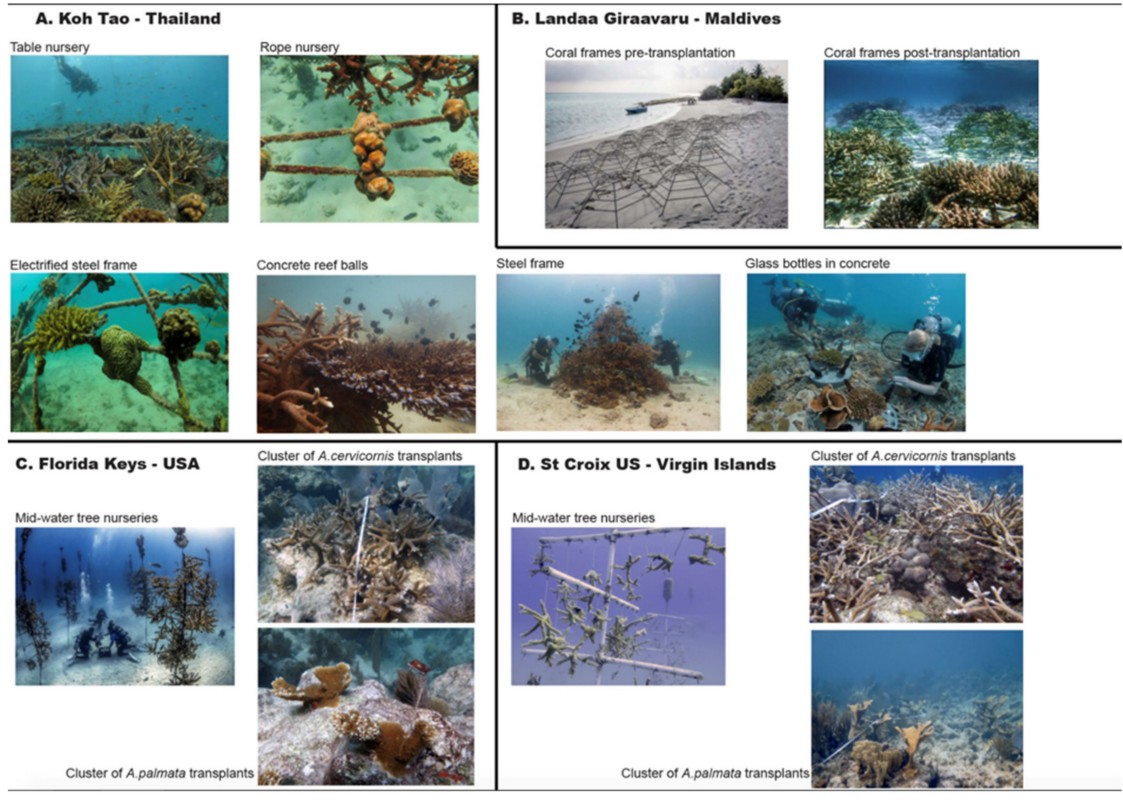

**Figure 2.** Photo montage illustrating coral restoration strategies at the four coral restoration programs surveyed. Photo credits to Margaux Hein, New Heaven Reef Conservation Program, Reefscapers and Marine Savers, and The Coral Restoration Foundation.

### 2.1.1. Box 1. Coral Restoration in Koh Tao, Thailand

Koh Tao is a moderately sized, high island (21 km$^2$ in area) located in the Gulf of Thailand. The island has undergone rapid development in the past 30 years and is now considered a global hotspot for SCUBA diving, with over 500,000 visitors every year [24]. This rapid development has been largely unregulated, and resorts, bars, and restaurants have replaced primary forests. What were once some of Thailand's most biodiverse and pristine reefs are now under stress from terrestrial run-off and sedimentation [25,26], over-use by the local water-based tourism industry [27,28], and both land-based and marine pollution [29]. Several studies have documented a high prevalence of coral disease and other indicators of compromised health [30–32]. Mass bleaching events recorded in 1998, 2010, and 2014 have also caused substantial coral mortality [33].

A restoration program led by the *New Heaven Reef Conservation Program* (NHRCP) was initiated in 2007 to assist the recovery of locally degraded reefs by re-building the complexity of coral assemblages, increasing coral cover, and alleviating diving pressures through widespread education. NHRCP uses a wide range of coral restoration techniques, from direct transplantation of coral fragments into natural holes and crevices on the reef to the building of artificial reef structures. Artificial structures are used preferentially in areas where the reef structure has been compromised by boat groundings and anchors, or smothered by sediment run-off from adjacent land. Types of structures used include steel frames, electrified artificial reefs, concrete reef balls, and glass bottles embedded in concrete (Figure 2A).

Corals are collected as fragments of opportunity, attached to mid-water ropes and table nurseries (Figure 2A) for a few months, and then attached onto the reef or onto one of the artificial structures. Attachment methods vary from epoxy cement to nylon thread, cable ties, or fine metal wire, depending on the type of structure. Restored areas are scattered around the island, and most include transplants attached to a variety of artificial reef structures, as well as directly onto the reef (Figure 1).

### 2.1.2. Box 2. Coral Restoration in Landaa Giraavaru, Maldives

Landaa Giraavaru is a small sand cay (0.18 km$^2$ in area) situated in Baa Atoll, a UNESCO Biosphere Reserve since 2011, on the western front of the Maldivian atoll chain. One five-star luxury resort, comprised of 23 individual villas, was built in 2004 and occupies the whole cay. Construction of the resort caused substantial structural damage to local reefs, which also suffered mass coral bleaching episodes and widespread coral mortality in 1998 and 2010 [34,35].

Coral restoration efforts led by the *Reefscapers* group (*Reefscapers Pvt Ltd*) primarily aim to increase biodiversity, reef complexity, and habitat diversity on the "house reef" surrounding the island. They use sand-coated steel structures, referred to as "coral frames", as artificial substrata on which to attach coral fragments. Three sizes of frames are used (small, medium, and large), ranging from 110 × 40 cm to 200 × 110 cm (width × height) (Figure 2B). Coral fragments are securely attached to frames with cable ties on land and the frames are then placed on the reef at depths ranging from 5 to 10 m around the island. As of March 2016, the reef around Landaa Giraavaru hosted 2800 frames, which covered an area of about 5500 m$^2$ and harbored 40 different species of corals (Figure 1). The first frames were populated with corals that were salvaged from the construction site when the resort was built in 2004. Nowadays, coral fragments are collected from colonies living on older frames, specifically targeting colonies that resisted earlier bleaching events.

### 2.1.3. Box 3. Coral Restoration in the Florida Keys, USA

The Florida Keys in the United States of America have a long history of disturbances that have resulted in dramatic loss of coral cover and diversity, particularly in the past 20 years [1,36,37]. Disturbances have included tropical storms (2005, 2008, 2012), coral bleaching associated with both cold-water anomalies (2010) and warm water anomalies (1987, 1997, 1998, 2005, 2014), and severe outbreaks of coral disease and of corallivores [37–40]. Like Koh Tao, the Florida Keys are a hotspot for reef-based tourism [41], and local reefs are thus suffering from a wide range of anthropogenic disturbances, including degraded water quality due to land-based sources of pollution [42], and high intensities of boating and diving activities [36].

The Coral Restoration Foundation (CRF) was created in 2007 with the specific objective of growing and restoring threatened species of corals in the genus *Acropora* (*A. cervicornis* and *A. palmata*). Abundances of these two species of corals have declined by up to 90% throughout the Caribbean and both have been listed as "critically endangered" since 2008 by the IUCN [23]. The Foundation harvests coral fragments from remnant colonies surviving on the reef and places them in coral nurseries. Early nursery prototypes included seafloor table nurseries, but they then developed coral tree nurseries that are suspended in the water column at approximately eight meters depth (Figure 2C). Once fragments are large enough, they are planted directly onto the reef substrata using a 2-part marine epoxy cement (Figure 2C). Restoration efforts extend over 31 sites on 10 reefs along the upper Florida Keys reef tract [23] (Figure 1).

### 2.1.4. Box 4. Coral Restoration in St Croix, US Virgin Islands

St Croix is a comparatively large high island (218 km$^2$ in area) forming part of the US Virgin Islands in the Caribbean. Reefs around St Croix have suffered extensively from climate change-related disturbances, similar to those described above for the Florida Keys. Tropical storms in 1989 and 1995 caused extensive reef damage, and several coral disease outbreaks over the past 20 years have caused further coral mortality [43,44]. In comparison to the Florida Keys, however, reefs around St Croix are not suffering from intense tourism pressure.

*The Nature Conservancy* (TNC) commenced coral restoration efforts in 2009, with the goal of growing and re-stocking endangered species of *Acropora* on local reefs [45]. Initially, corals were collected as fragments of opportunity that had been broken from parent colonies naturally by storm or surge events. Currently, fragments are collected from donor colonies and grown in coral tree nurseries,

following methods developed by CRF in Florida. Once fragments are large enough, they are planted back onto the reef using a 2-part marine epoxy cement. Restoration sites are scattered around the Island, with a particular focus on *A. cervicornis* restoration on the North Shore reefs of Cane Bay, and on *A. palmata* restoration near Green Cay and Knights Bay (Figures 1 and 2D).

*2.2. Measuring 'Snap Shots' of Coral Assemblages' Response to Restoration*

At each of the four locations, benthic data were compared among replicate restored sites (R), unrestored sites (UR), and control reference sites (CR). Sites were carefully selected with local reef managers to ensure we had the best representation for all three categories. At restored sites, coral fragments had been transplanted either directly onto the substrata or onto artificial structures. Unrestored sites were degraded sites directly adjacent to restored sites but were not the subject of coral restoration efforts. Unrestored sites were used as the direct control against which to assess potential effects of the restoration effort. Control reference sites were comparatively undisturbed sites nearby that were exposed to similar environmental conditions, thus their reef communities were hypothesized to be similar to those at the R and UR sites prior to degradation. CR sites acted as an indirect control providing some additional reference of the natural variability in reef condition at each particular location. A minimum of three replicate sites were surveyed for each of the three treatments (R, UR, CR) at each location, except at St Croix, where the extent of appropriate undisturbed reef area was so small that we could only survey two control reference sites. Thus, three restored sites, three unrestored sites, and three healthy reference sites were surveyed at all locations (except for the two CR sites at St Croix). In addition, a fourth restored site and a fourth unrestored site were surveyed in St Croix.

Benthic data were recorded along three 20-m transect lines at each of the three sites per treatment in Koh Tao, Landaa Giraavaru, and the Florida Keys, for a total of 180 m surveyed per treatment at each of these locations. In St Croix, the restored area was too small for three replicate 20-m transects, thus two replicate 22.5-m transects were surveyed at each of four R and four UR sites (i.e., 180 m surveyed per treatment) to match the overall transect lengths surveyed at other locations.

2.2.1. Benthic Cover and Structural Complexity

Benthic cover was measured using the line-intercept method, whereby the length of each substrate category falling directly under the line was recorded to the nearest centimeter [46]. Substrate categories included all scleractinian corals, which were identified to the genus level; soft corals; macro-algae; other substrata like sand, rubble, and rocks; and other organisms, such as sponges, corallimorphs, and zoanthids. Percent cover of each substrate category was then calculated relative to the total length of each transect.

Structural complexity of the reef under each transect line was scored qualitatively using a scale from 0 to 5, where 0 = no relief, and 5 = high structural complexity and high coral cover, following methods described in [20,47].

2.2.2. Coral Health, Generic Richness, and Juvenile Recruitment

In addition to line-intercept surveys of coral cover, 2-m-wide belts were surveyed along each transect line (i.e., a 40-m$^2$ area per transect), within which all hard corals were identified to the genus level and assigned to a coral health category. Corals were scored as either healthy or having signs of one or more disease types, and/or a range of compromised health states, such as algal overgrowth, sediment smothering, physical damage, or signs of predation. The prevalence of each disease or compromised health category was calculated as its percentage relative to the total number of coral colonies surveyed in each 40-m$^2$ belt transect. Coral health categories and assessment protocols followed guidelines developed for the Indo-Pacific [48], and for Caribbean reefs [49]. These survey techniques have been applied previously to assess coral health (e.g., [31,50]). The number of coral genera recorded in each belt transect was used as a measure of generic richness. The number of coral

juveniles (colonies with a diameter under 5 cm; [51]) was also recorded within each belt transect, and used as a proxy for the number of coral recruits in recent years [52].

*2.3. Data Analysis*

All data were analyzed using the statistical program R (version 3.4.1, [53]). The analyses described below were applied to metrics measured at each of the four locations separately. Given the large geographic distances between the four locations and inherent differences in biodiversity and coral cover among their reef communities, only qualitative comparisons of the summative results are made among the four reef locations.

2.3.1. Benthic Cover

For each of the four locations, the mean percent cover of each substrate category was compared among treatments (R, UR, and CR) and sites (n = 3 or 4 sites per treatment type) using multi-factor general linear models (GLMs). Treatments were analyzed as fixed factors and sites as random factors. A variety of models were tested, including ones where explanatory variables were treated as having either additive or multiplicative effects, and where data were log-transformed. AICc model selection was used to select the model explaining the greatest variation in the data, i.e., the model having the lowest AICc score. Assumptions for model validity were checked through Q-Q plots and residual plots. When tests failed to meet the assumptions of a Gaussian distribution after log-transformation, non-parametric Kruskal–Wallis tests were applied. When applicable, post-hoc Tukey's HSD tests were also applied to tease out differences among treatments and sites.

2.3.2. Structural Complexity

Analyses of the mean structural complexity scores among treatments and sites at each location were performed using multi-factor general linear models, as described above for benthic cover analyses.

2.3.3. Coral Generic Richness Abundance of Juvenile Corals

Multi-factor general linear models were also used to compare generic richness and abundance of juvenile corals among treatments and sites at each location. Details of analyses and checks of assumptions were as described above for benthic cover data, except that data were modelled as having "Poisson" or "negative binomial" distributions, as these are the most appropriate distributions for count data. Analysis of coral juvenile abundance could only be done for two of the four sites: Koh Tao and Landaa Giraavaru, as sites in the Florida Keys and St Croix were data deficient for this indicator, i.e., there was little to no recruitment at any of these sites.

2.3.4. Coral Health

Percentages of corals in each health category were compared among treatments and sites using analyses similar to those described above for benthic cover. Prevalence values for each of four health categories were compared among treatments and sites at each location, namely the prevalence of healthy corals, diseased corals, corals with signs of predation, and corals with other signs of compromised health.

2.3.5. Coral Assemblages

Multivariate analyses were used to assess potential differences in the composition of coral assemblages (i.e., abundance of local coral genera and other benthic categories) among treatments at each location. Prior to analysis, all data were transformed using Wisconsin's double transformation for the fourth root. We then created distance matrices based on "Bray-Curtis" dissimilarity indices, as these are good at detecting ecological gradients [54], and applied non-metric multidimensional scaling (nMDS) to the transformed dataset. The validity of the nMDS was checked through evaluation

of the $R^2$ value of the linear and non-linear fit, as well as the stress value, which was assumed to be good when <0.2 [55]. Coral health and benthic cover data were overlaid on top of the nMDS, and ADONIS tests (multivariate ANOVA based on dissimilarities) were used to calculate the contribution of each variable to the spread of the benthic community data, as well as to the differences in the composition of coral assemblages between sites at each location (pairwise ADONIS). Finally, SIMPER analyses were performed to reveal the cumulative contributions of the most influential coral genera and benthic category to the spread of the data at each location.

## 3. Results

### 3.1. Hard Coral Cover

Mean hard coral cover was more than twice as great at restored treatments compared to degraded unrestored treatments at three of the four locations: Koh Tao (LM: F = 9.5 $p < 0.001$, Table S1), Landaa Giraavaru (LM: F = 6.9, $p < 0.001$, Table S1), and the Florida Keys (GLM: Residual Deviance = 9.4, $p = 0.005$, Table S1, Figure 3). In St Croix, there was a trend towards higher hard coral cover at restored treatments compared to unrestored treatments, but the difference was not statistically significant (GLM: RD = 1.7, $p = 0.375$, Table S1, Figure 3).

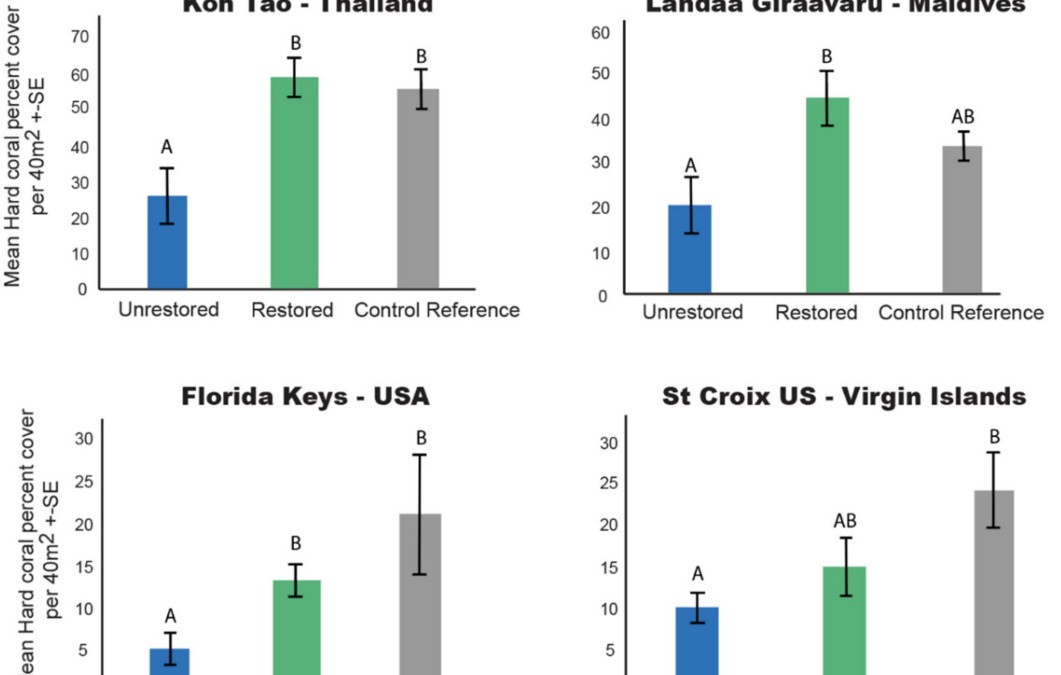

**Figure 3.** Mean percent cover of hard corals per 40-$m^2$ belt transect (±SE) compared among treatments (unrestored, restored, control reference sites) at each of the four locations. Letters above each histogram indicate whether mean values differ significantly (different letters) or are statistically indistinguishable (same letters). n = 9 transects per treatment in Koh Tao, Landaa Giraavaru, and the Florida Keys; In St Croix, n = 8 transects for unrestored and restored treatments, n = 6 transects for the control reference treatment.

There were trends for absolute values of mean hard coral cover to be higher at restored treatments than at control reference treatments at the two Indo-Pacific locations (Koh Tao and Landaa Giraavaru); conversely, means were highest at control reference treatments at both Caribbean locations (Florida and St Croix; Figure 3). However, at all four locations, differences in mean hard coral cover between restored treatments and control reference treatments were not statistically significant (Figure 3, Table S1).



### 3.2. Structural Complexity

Structural complexity of the coral community was significantly higher at restored treatments compared to unrestored degraded treatments at all four locations (Figure 4, Table S2). In Koh Tao, structural complexity scores were 2 times greater at restored compared to unrestored treatments (LM: F = 23.18, $p < 0.001$, Table S2), and 1.5 times greater at restored compared to control reference treatments (GLM: $p = 0.0013$, Table S2). At all three other locations, although structural complexity scores were 1.5 times greater at restored than at unrestored treatments (Landaa Giraavaru LM: F = 6.9, $p = 0.0014$, Florida Keys LM, F = 11.5, $p = 0.019$, St Croix LM, F = 19.4, $p < 0.001$, Table S2), and mean scores were highest at control reference treatments (although not significantly different; Figure 4). Scores at restored treatments were consistently greater than the average score across all sites (2.5 out of 5), whereas scores at unrestored treatments were consistently lower (Figure 4).

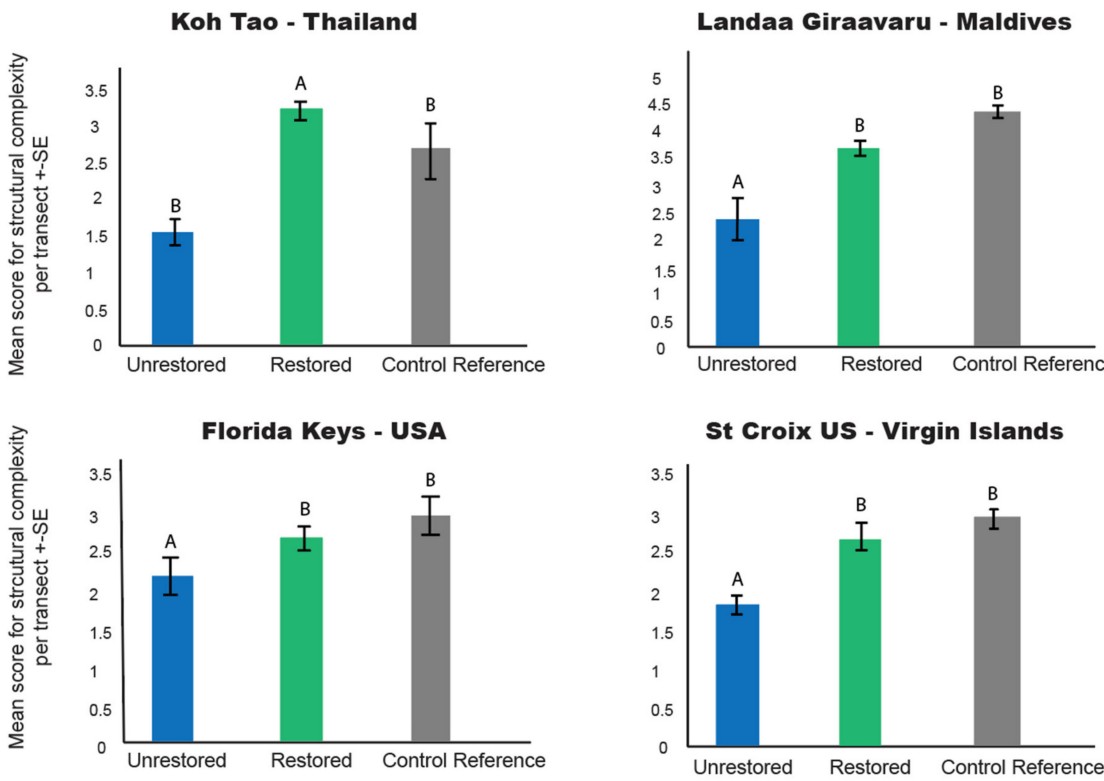

**Figure 4.** Mean structural complexity scores (±SE) compared among treatments (unrestored, restored, control reference sites) at each of the four locations. Letters above each histogram indicate whether mean values differ significantly (different letters) or are statistically indistinguishable (same letters). n = 9 transects per treatment in Koh Tao, Landaa Giraavaru, and the Florida Keys; in St Croix, n = 8 transects for unrestored and restored treatments, n = 6 transects for the control reference treatment.

### 3.3. Number of Coral Juveniles

This indicator was only valid for Koh Tao and Landaa Giraavaru because juvenile coral colonies were not detected in high enough abundance in the Florida Keys or St Croix to provide sufficient data for statistical analyses.

In Koh Tao, the mean abundance of juvenile corals was greatest at restored treatments. Although abundances at restored treatments were not significantly greater than those at control reference treatments (Table S3, Figure 5), they were significantly greater than those at unrestored treatments, where no juveniles were recorded (Kruskal–Wallis: $\chi^2 = 8.22$, df = 2, $p = 0.043$, Table S3, Figure 5). Overall, the mean number of juveniles recorded in Koh Tao was 5.7/40 m$^2$, with abundances differing among restored treatments according to the artificial structures used (Kruskal–Wallis: $\chi^2 = 6.06$,

df = 2, *p* = 0.049, Table S4). The highest number of juveniles recorded were on concrete reef balls in Tanote Bay (Figure 6), and the lowest number recruited to the mix of steel frames and bottle nurseries in Chalok Bay (Figure 6). In Landaa Giraavaru, the mean abundance of coral juveniles (8 juveniles/40 m$^2$ across all sites and treatments) did not significantly differ among the three treatments (Kruskal–Wallis: $\chi^2$ = 0.825, df = 2, *p* = 0.66; Figure 5, Table S3).

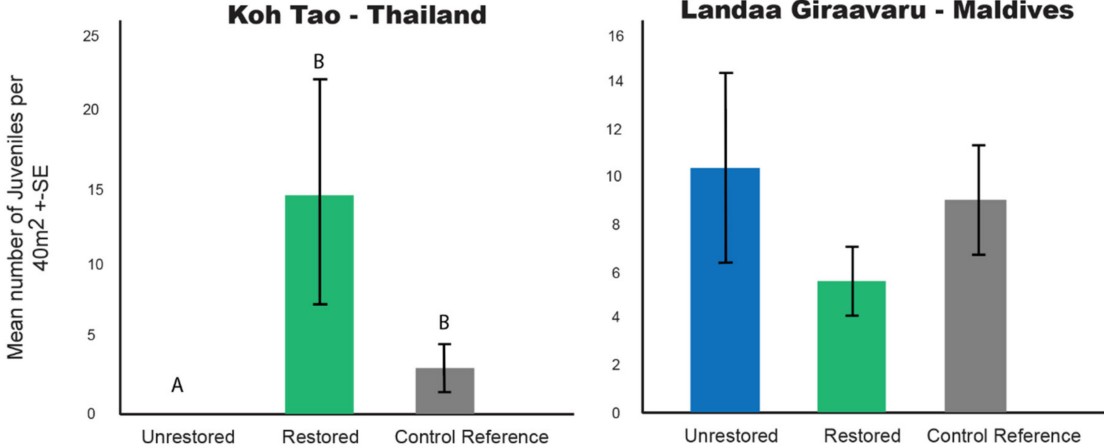

**Figure 5.** Mean number of juvenile corals counted per 40-m$^2$ belt transect (±SE) compared among treatments (unrestored, restored, control reference sites) in Koh Tao (Thailand) and Landaa Giraavaru (Maldives). Letters above each histogram indicate whether mean values differ significantly (different letters) or are statistically indistinguishable (same letters). n = 9 transects per treatment.

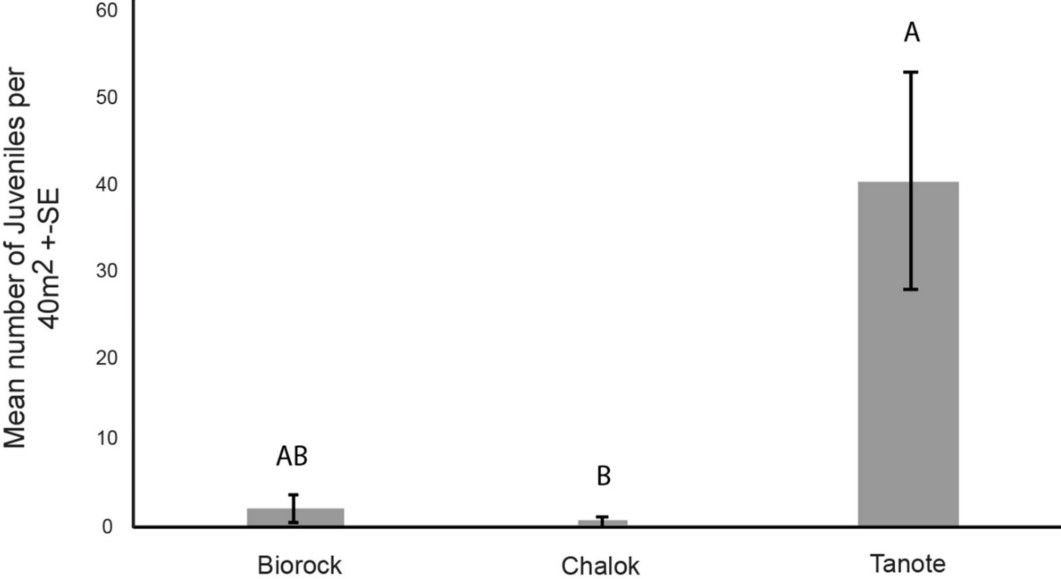

**Figure 6.** Mean number of juvenile corals counted per 40-m$^2$ belt transect (±SE) compared among the three restored sites in Koh Tao (Thailand). Restoration designs varied among the three sites, i.e., corals were transplanted: onto electrified steel frames at the Biorock site, onto steel frames and into glass bottles embedded in concrete in Chalok, and onto concrete reef balls in Tanote. Letters above each histogram indicate whether mean values differ significantly (different letters) or are statistically indistinguishable (same letters). n = 9 transects per treatment.

### 3.4. Coral Generic Richness

Increases in coral generic richness at restored compared to unrestored treatments only occurred in Koh Tao (GLM: RD = 13.2, *p* = 0.0352, Table S5, Figure 7). In both the Florida Keys and St Croix,

coral generic richness was similar across all treatments at all locations (Table S5, Figure 7). In Landaa Giraavaru, coral generic richness was significantly lower at the restored treatments compared to both unrestored (GLM: RD = 29.2, *p* = 0.0015) and control reference treatments (GLM: RD = 29.2, *p* < 0.001), (Table S5, Figure 7).

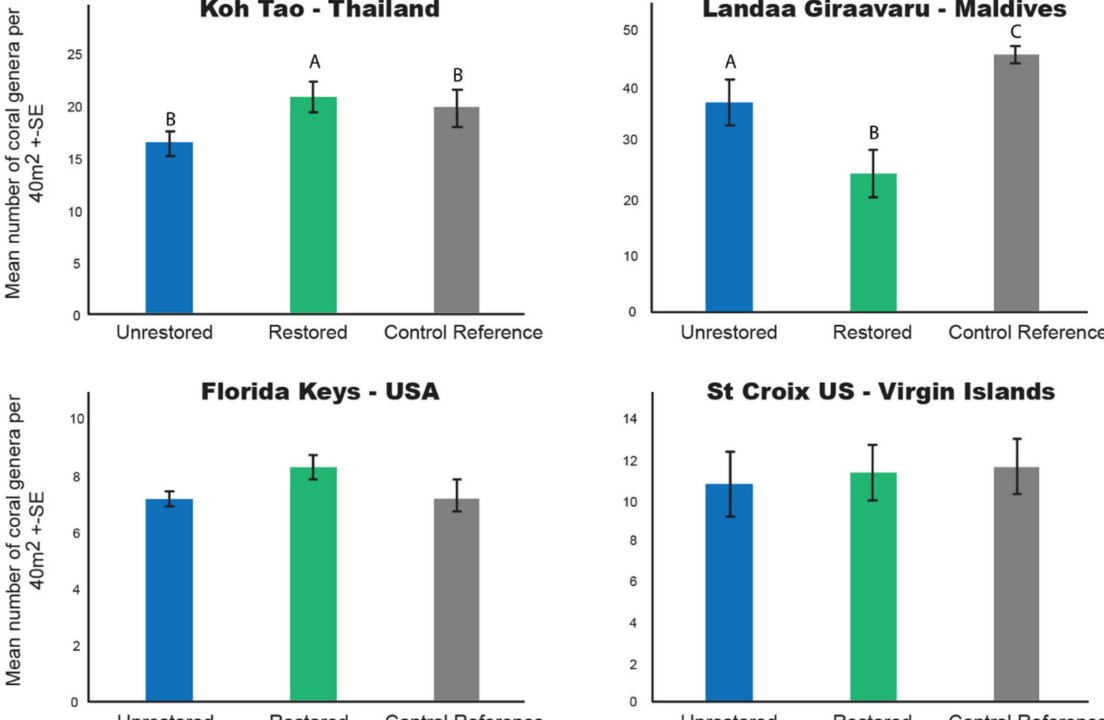

**Figure 7.** Mean number of coral genera per 40-m$^2$ belt transect (±SE) compared among treatments (unrestored, restored, control reference sites) at each of the four locations. Letters above each histogram indicate whether mean values differ significantly (different letters) or are statistically indistinguishable (same letters). n = 9 transects per treatment in Koh Tao, Landaa Giraavaru, and the Florida Keys; in St Croix, n = 8 transects for unrestored and restored treatments, n = 6 transects for the control reference treatment.

### 3.5. Coral Health

Coral health varied among treatments and locations. In Koh Tao, unrestored treatments had a four-fold higher prevalence of unhealthy coral colonies compared to both restored and control reference treatments (GLM: RD = 1534, *p* < 0.001, Table S6), which was driven by a four-fold higher prevalence of coral colonies with signs of compromised health (GLM: RD = 4.35, *p* < 0.001, Table S8, Figure 8). The prevalence of diseased corals and of colonies with signs of predation did not differ among treatments (Figure 8, Table S7). Signs of predation in Koh Tao were primarily identified as feeding scars from *Drupella* snails and crown-of-thorns starfish (COTS).

In Landaa Giraavaru, the prevalence of unhealthy coral colonies was consistently over 80% of all colonies in all treatments. The overall high prevalence of unhealthy corals was driven by a high (62.4%) mean prevalence of bleached corals. Disease prevalence was also two times greater at restored compared to control reference treatments (GLM: RD = 6.03, *p* = 0.025, Figure 8, Table S7).

In the Florida Keys, disease prevalence was highest at control reference treatments; disease prevalence was 1.5 times greater than at restored sites (GLM: RD = 1.64, *p* = 0.028, Table S7), and 2.8 times greater than at unrestored treatments (GLM: RD = 1.64, *p* = 0.006, Table S7, Figure 8). Only restored treatments had signs of predation, thus the prevalence of predation scars was significantly higher at these treatments compared to both unrestored (Kruskal–Wallis, Chi-square = 21.034, df = 2,

$p = 0.038$, Table S9) and control reference treatments (Kruskal–Wallis, Chi-square = 21.034, df = 2, $p = 0.038$, Table S9, Figure 8).

In St Croix, restored treatments had a higher prevalence of diseased colonies than unrestored (GLM: RD = 0.41, $p < 0.001$, Table S7) and control reference treatments (GLM: RD = 0.41, $p = 0.037$, Table S7), and a higher prevalence of colonies with other signs of compromised health than control reference treatments (GLM, RD = 0.92, $p < 0.001$, Table S8, Figure 8). Restored treatments were also the only sites at which we observed signs of predation (Figure 8). Signs of predation in both the Florida Keys and St Croix were dominated by scars from flatworms and fish bites.

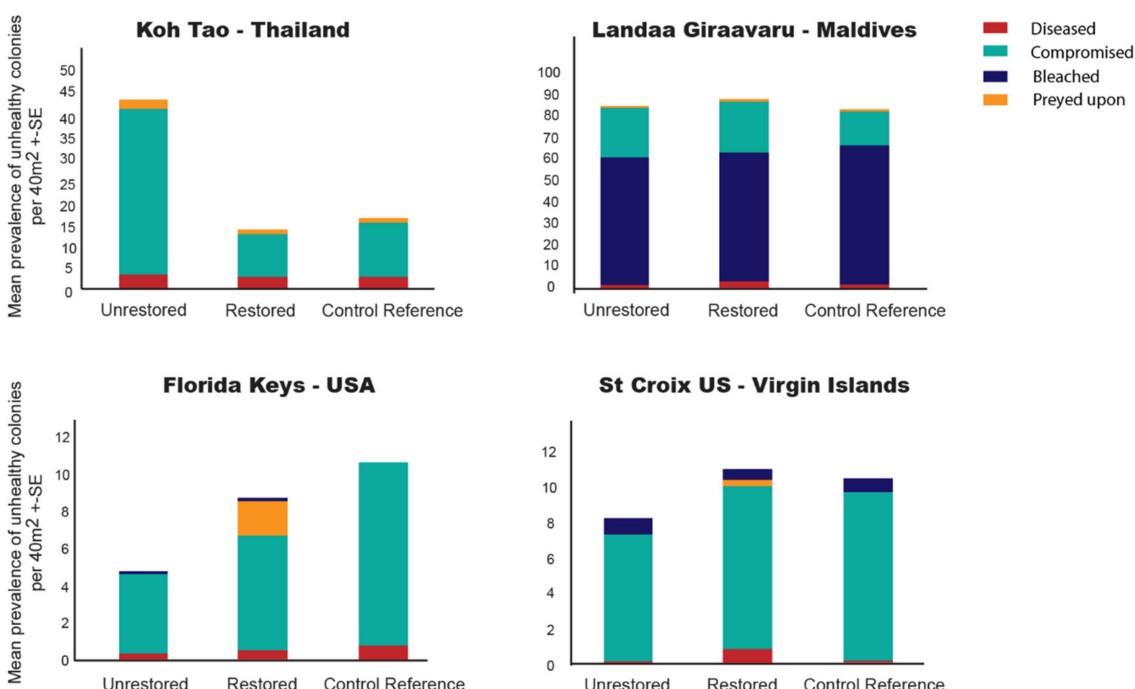

**Figure 8.** Mean prevalence of corals in four health categories representing unhealthy states (corals with signs of disease, bleaching, predation, or other signs of compromised health) per 40-m$^2$ belt transect compared among treatments (unrestored, restored, control reference sites) at each of the four locations. n = 9 transects per treatment in Koh Tao, Landaa Giraavaru, and the Florida Keys; In St Croix, n = 8 transects for unrestored and restored treatments, n = 6 transects for the control reference treatment.

### 3.6. Composition of the Coral Assemblages

The composition of coral assemblages differed among restoration treatments at all four locations. In Koh Tao, the composition of coral assemblages at control reference treatments differed significantly from those at both restored and unrestored treatments (ADONIS: (CR to R) F = 3.64, $p = 0.014$; (CR to UR) F = 4.52, $p = 0.008$, Table S10, Figure 9). There was also a significant effect of sites on the composition of the coral assemblages (ADONIS: F = 5.67, $p = 0.001$). Overall, coral assemblage composition at the restored treatments was intermediate between those at the unrestored and control reference treatments (Figure 9). Restored treatments had four times more cover of corals in the family Acroporidae than either the unrestored or control reference treatments (Figure 10). Accordingly, the cumulative contribution of Acroporidae accounted for 75% of the differences between restored and unrestored treatments (SIMPER). Sand dominated the benthos at unrestored treatments, accounting for 47% (SIMPER, cumulative contributions) of the differences between unrestored and restored treatments, and 38% of the differences between unrestored and control reference treatments (SIMPER, cumulative contributions). Poritidae and Fungiidae were also more abundant at control reference treatments than restored and unrestored treatments (Figure 10).

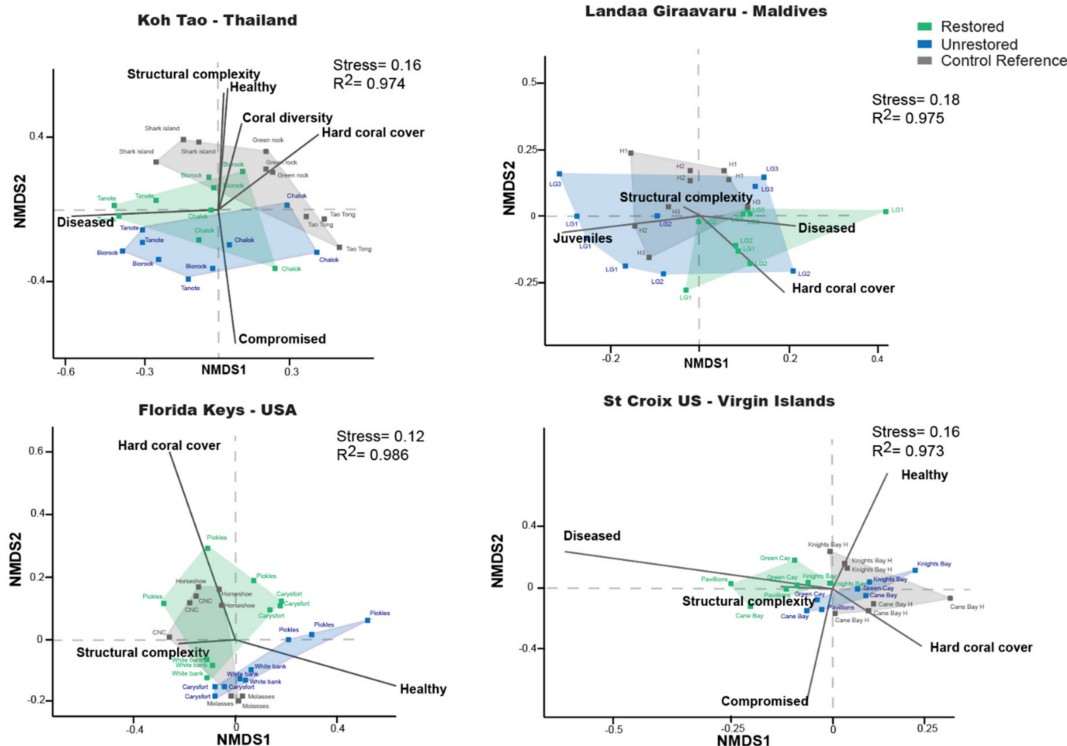

**Figure 9.** Effect of coral restoration on the composition of coral assemblages at four geographic locations, as represented by non-metric multidimensional scaling. Polygons represent coral assemblages in each treatment, where green polygons encompass restored sites, blue polygons encompass unrestored sites, and grey polygons encompass control reference sites. Colored shading reflects the location of the respective set of sites in non-metric multi-dimensional scaling space. Vectors represent the influence of benthic attributes on the benthic community composition.

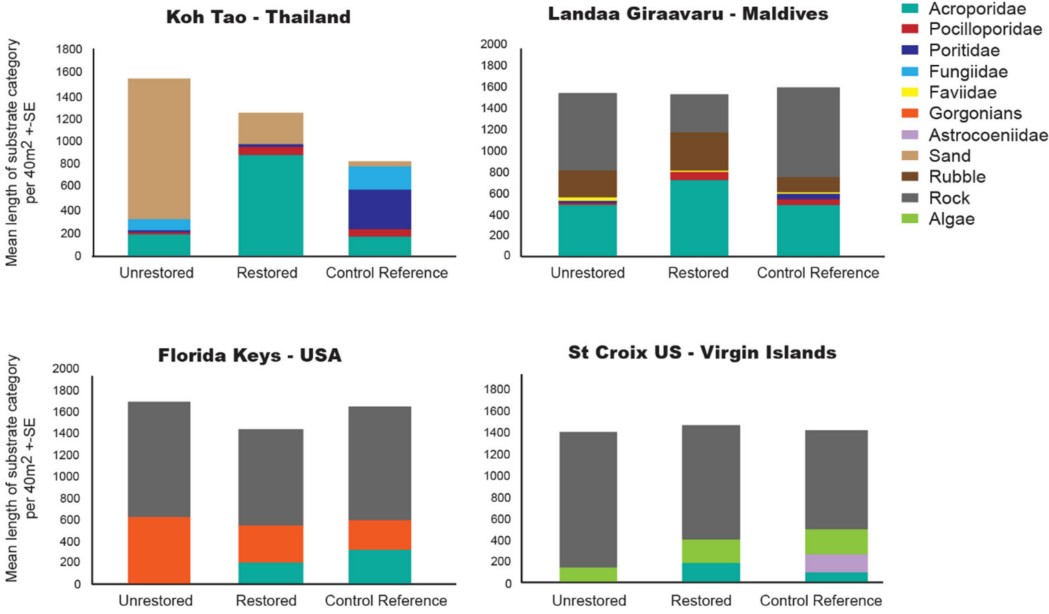

**Figure 10.** Comparisons of the mean cover of the most influential substrate categories (post-simper analyses) per 40-m$^2$ belt transect among treatments (unrestored, restored, reference control sites) at each of the four locations. n = 9 transects per treatment in Koh Tao, Landaa Giraavaru, and the Florida Keys; In St Croix, n = 8 transects for unrestored and restored treatments, n = 6 transects for the control reference treatment.

In Landaa Giraavaru, the composition of coral assemblages at the restored treatments differed significantly from the composition of assemblages at unrestored and control treatments (ADONIS: (R to UR) F = 3.33, *p* = 0.15; (R to CR) F = 3.78, *p* = 0.005, Table S10). Coral assemblages were also significantly different at control compared to unrestored treatments (ADONIS: F = 2.29, *p* = 0.045, Table S10). There was also a significant effect of site on the composition of coral assemblages (ADONIS: F = 2.18, *p* = 0.004). Restored treatments were characterized by higher cover of corals in the family Acroporidae and by lower cover of rubble (Figure 10). Rubble contributed to 30% of the differences between restored and unrestored treatments, and 72% of the differences between restored and control reference treatments (SIMPER, cumulative contributions). Acroporidae contributed 58% of the differences between restored and unrestored treatments, and 55% of the differences between restored and control reference treatments (SIMPER, cumulative contributions).

In the Florida Keys, only unrestored treatments had a distinct benthic community composition (ADONIS: (UR to R) F = 3.52, *p* = 0.014; (UR to CR) F = 3.88, *p* = 0.006, Table S10, Figure 9). There was also a significant site effect on the composition of the benthic community (ADONIS: F = 3.88, *p* = 0.001). In terms of benthic composition, rocks, gorgonians, and Acroporidae were the most influential factors driving differences among treatments (SIMPER). The cover of corals in the family Acroporidae was nil at unrestored treatments, and highest at control reference treatments. Acroporidae accounted for 56% of the differences between unrestored and control treatments, 84% of the differences between unrestored and restored treatments (SIMPER, cumulative contribution), and 64% between restored and control reference treatments (SIMPER, cumulative contribution) (Figure 10). Rocky substrate and gorgonians had the highest percent cover in unrestored treatments (Figure 10). Rocky substrate accounted for 32% of the differences between unrestored and restored treatments, and 80% of the differences between unrestored and control reference treatments (SIMPER, cumulative contribution). Gorgonian cover was twice as high in unrestored compared to both restored and control reference treatments and thus accounted for 65% of the differences between unrestored and restored treatments, and 29% of the differences between unrestored and control reference treatments (SIMPER, cumulative contribution) (Figure 10).

In St Croix, the coral assemblages at restored treatments differed significantly from those of both unrestored and control reference treatments (ADONIS: (R to UR) F = 6.96, *p* = 0.001; (R to CR) F = 3.5, *p* = 0.004, Table S10). The coral assemblages at control reference treatments were also distinct from those of the unrestored treatments (ADONIS: F = 3.15, *p* = 0.017, Table S10). Benthic community composition also varied significantly among sites (ADONIS: F = 3.49, *p* = 0.001). Differences in benthic community composition were driven by the cover of corals in the family Acroporidae, which was 1.9 times greater at restored than at control reference treatments; Acroporidae corals were absent at unrestored treatments (Figure 10). Acroporidae therefore accounted for 71% of the differences between unrestored and restored treatments (SIMPER, cumulative contribution). Astrocoeniidae were only present in control reference treatments and accounted for respectively 69% and 65% of the differences in the benthic community between restored and control reference treatments, and between unrestored and control reference treatments (SIMPER, cumulative contribution). Benthic communities at unrestored treatments were dominated by the presence of rocks and algae (Figure 10).

The effects of benthic attributes on the composition of coral assemblages at all four locations is further explored in the Supplementary Material (Section S2).

### 3.7. Summary and Links with Restoration Designs

The effect of coral restoration on the five ecological indicators surveyed differed among the four study locations, reflecting differences in restoration designs and local factors (Figure 11). While our snapshot surveys prevent us from linking restoration outcomes to specific designs, some designs may have warranted stronger responses than others. For example, all five indicators surveyed positively increased in restored treatments in Koh Tao, where the restoration design includes a mix of direct transplantation and a variety of artificial structures (steel frames, electrified steel frames, concrete reef

balls, and glass bottles in concrete). This combination of techniques led to the highest rate of increase in structural complexity, coral generic diversity, number of juveniles, and improved coral health at restored compared to unrestored treatments of all study locations (Figure 11). Other designs that used artificial structures like the steel frames in Landaa Giraavaru also led to significant increases in hard coral cover and structural complexity at restored compared to unrestored treatments (Figure 11), but restoration outcomes at this location also included significant decreases in coral generic richness at restored treatments. Direct transplantation was the only technique used in both the Florida Keys and St Croix. This technique resulted in consistent increases in hard coral cover, structural complexity, and coral generic diversity (Figure 11). In the Florida Keys, the restoration design also led to five times greater hard coral cover at restored compared to unrestored treatments, thus this metric increased by the greatest amount in the Florida Keys out of all four study locations (Figure 11). Conversely, increases in hard coral cover at restored compared to unrestored treatments were the lowest in St Croix (Figure 11). Finally, coral health was poorer in restored compared to unrestored treatments in both the Florida Keys and St Croix.

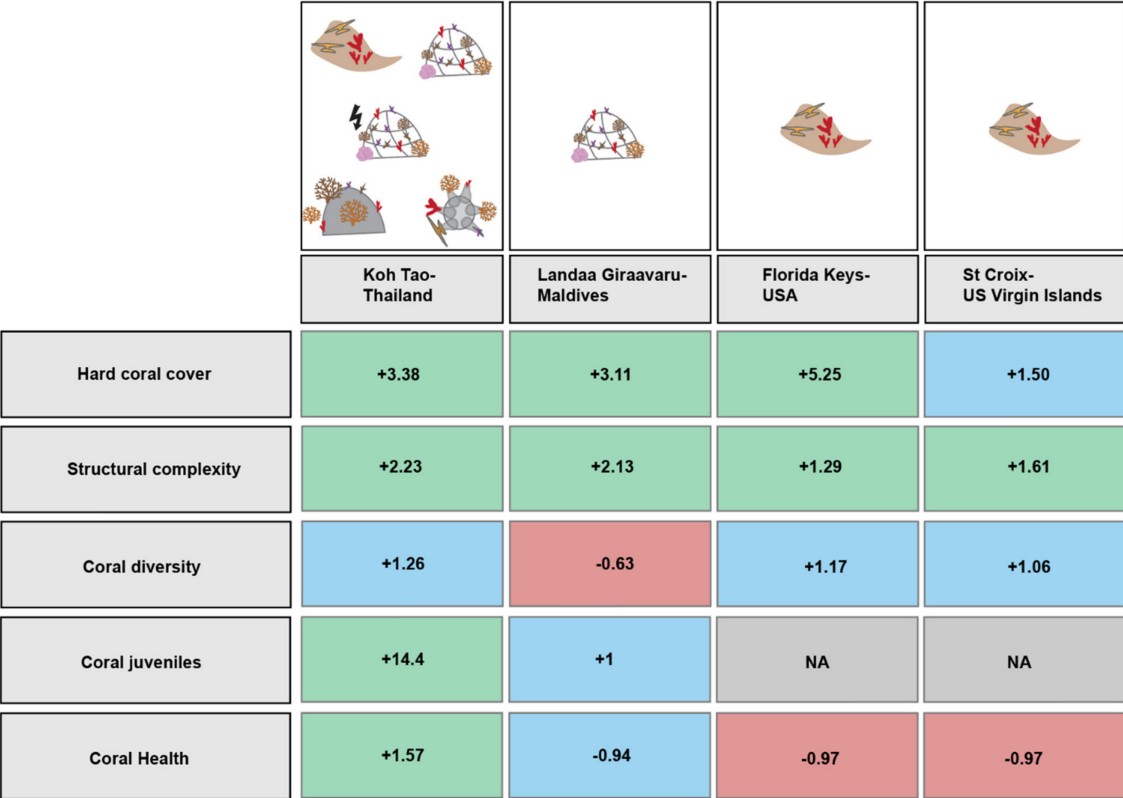

**Figure 11.** Summary table comparing five ecological indicators surveyed at four study locations with different restoration designs. Numerical values represent the ratios of each metric at restored compared to unrestored treatments. Colored boxes represent the significance of the difference between restored and unrestored treatments. Green denotes significant positive ratios; red denotes significant negative ratios; blue denotes non-significant differences.

## 4. Discussion

This study is the first to examine the long-term effects of coral restoration practices on coral assemblages and to test the generality of outcomes across programs using differing protocols in a range of geographic locations. We found systematic increases in hard coral cover and reef structural complexity at restored compared to unrestored treatments at all four locations surveyed. Moreover, multivariate analyses confirmed that outplanted corals had substantial impacts on local benthic communities, resulting in the community composition at restored sites differing from that of unrestored

and comparatively healthy control sites. Patterns in the responses of other ecological indicators of reef resilience to restoration varied across locations, potentially reflecting variations in local benthic assemblages and/or variations in community responses to different restoration methodologies.

### 4.1. Restoration Increases Coral Cover and Structural Complexity

The doubling of hard coral cover at restored compared to unrestored treatments at three out of four locations indicates that the range of restoration techniques investigated here are effective strategies for restoring coral assemblages. Moreover, coral cover was higher in restored plots than at control reference treatments following 10 years of restoration at both Indo-Pacific locations (Koh Tao and Landaa Giraavaru). While coral cover remained highest at control reference treatments in the Florida Keys and St Croix, the restoration goals of these two Caribbean programs were more focused on growing and restoring endangered species of *Acropora* (*A. cervicornis* and *A. palmata*) [23]. Systematic increases in hard coral cover at restored treatments are unsurprising, as corals fragments were actively planted at all four locations. However, results suggest that while corals may suffer post-transplant stress and mortality [56–58], restoration efforts at all four locations are substantive enough to have positive effects on coral cover over 10-year timeframes. Increased hard coral cover is a necessary first step towards increasing reef resilience, increasing local breeding populations of corals, providing habitats for juvenile fish and invertebrates, and potentially preventing or at least mitigating phase shifts towards algae-dominated systems [1,59].

Significant increases in structural complexity at restored compared to unrestored treatments at all four study locations suggest that both direct transplantation of coral fragments onto the reef substrata and transplantation onto artificial structures are effective in increasing reef complexity at restored sites. In Koh Tao, where coral fragments are generally attached to artificial structures, structural complexity was doubled at restored compared to unrestored treatments, and higher at restored compared to control reference treatments. Although artificial structures were used in Landaa Giraavaru, structural complexity did not differ significantly between restored and reference treatments, largely because of the high natural complexity of control reference reefs (mean structural complexity greater than 4 out of 5). Here, complexity represents the degree of reef relief (cf. [47]) but does not specifically account for the number and sizes of holes and crevices present in the reef matrix, which may affect the abundance and diversity of fish and invertebrates [60]. In the Florida Keys and St Croix, a lack of difference in structural complexity between restored and control reference treatments reflects the fact that most of the complexity at these locations is provided by the presence or absence of thickets of branching *Acropora*, which are the targets of the restoration efforts [61].

### 4.2. Other Coral-Based Indicators of Reef Resilience Vary among Restoration Programs

Despite increases in coral cover and structural complexity at restored treatments, other critical indicators of reef resilience did not increase consistently in response to restoration efforts. For example, higher densities of juvenile corals at restored compared to unrestored treatments were only found in Koh Tao, and only on concrete reef balls. It may be that the high surface rugosity of reef balls is conducive to coral larval settlement [62,63]. However, because Koh Tao was the only restoration program out of the four studied to use these structures, and they were only used at one out of the three restored sites, we are unable to distinguish between the potential contributions of site versus type of structure on the increased abundance of coral juveniles at this one site.

In Landaa Giraavaru, lack of differences in juvenile coral density among treatments might be attributable to either the type of structure used (i.e., steel frame structures that are not conducive to larval settlement), and/or the fact that reefs around the island are limited by recruitment. Here, the average number of juveniles recorded across all sites ($0.8/m^2$) was much lower than coral recruit densities previously reported in the Maldives (2.5 to 18 recruits/$m^2$, [57]), and in other regions of the world (4 to 80 recruits/$m^2$, [64,65]). However, these studies define coral recruits as any new corals colonizing the restored sites [62], and use different survey techniques (e.g., recruitment tiles, [65]). It is possible that

the methods used here, of only recording corals with a diameter <5 cm in 2-m belt transects, may have limited the detection of coral recruits. This interpretation is supported by findings of similar densities of recruits on Lord Howe Island reefs using the same methods [52]. The paucity of recruitment in both the Florida Keys and St Croix precluded investigating the effect of coral restoration on the abundance of juvenile corals at these two locations, and further confirms that reefs in the Caribbean are severely limited in their ability to recruit new juvenile corals [66,67].

Coral generic richness was a second indicator of reef resilience that was not consistently augmented by restoration programs. Coral restoration only positively affected coral generic richness in Koh Tao, where the restoration design explicitly aims to maximize the diversity of coral transplants. In the three other locations, targeted transplantation of specific corals meant that coral generic diversity was either lowest at the restored treatments (Landaa Giraavaru) or indistinguishable from unrestored treatments (Florida Keys, St Croix). In Landaa Giraavaru, coral transplants were dominated by fast-growing branching corals from the genera *Acropora* and *Pocillopora*, artificially boosting the density of these two genera at restored sites. The lack of a restoration effect on the generic richness in the Florida Keys and St Croix was unsurprising given that restoration efforts target the two endangered species of *Acropora* [23].

Finally, coral health, a third indicator of reef resilience that was not consistently improved by restoration, revealed location-specific patterns. Again, this indicator was improved only in Koh Tao, potentially because of the high level of maintenance of the restoration sites by the NHRCP team. It is also likely that elevation of the corals slightly above the substrata on artificial structures prevented them from being smothered by sediments or algae. Unrestored treatments had a significantly higher prevalence of colonies with sediment damage and algal overgrowth (included in the category 'other signs of compromised health'), corroborating this line of reasoning. It is noteworthy that there was no evidence that transplanted fragments were more susceptible to disease due to manipulation and injuries sustained in the process of attaching them to structures. In summary, results from Koh Tao suggest that planting corals above the substrata and maximizing the diversity of corals transplanted are good strategies to maximize coral health at restored sites.

In Landaa Giraavaru, poor coral health in all treatments reflected that, at the time of the survey, the Maldives were experiencing mass coral bleaching. Corals at all survey locations were severely bleached regardless of the depth or restoration treatment. The overriding impact of thermal stress at the time of the surveys is a reminder that active intervention approaches like coral restoration cannot prevent climate-driven exposure events that overwhelm reef resilience.

In the Florida Keys, coral disease prevalence was highest at control reference treatments, potentially because of high densities of *Acropora* combined with no active maintenance of natural reef areas, and the overall history of disease-related loss of Caribbean species of *Acropora* [39,68]. The prevalence of predation scars, on the other hand, was highest at the restored treatments, likely reflecting fireworm predation on freshly planted *A. cervicornis* [61,69,70].

In St Croix, restored treatments were again the only sites to experience coral predation at that location. Together with higher disease prevalence, restored treatments had overall lower coral health than either unrestored or control reference treatments. Results from both the Florida Keys and St Croix raise questions about whether Acroporids are good candidates for coral restoration in the Caribbean. While the two Caribbean programs are meeting their goal of increasing *Acropora* cover at restored sites [71], focusing on this genus alone might not lead to successful long-term outcomes in terms of reef resilience and enhanced reef-related ecosystem goods and services. Maximizing the diversity of coral transplants at these locations might help harness natural ecological processes that decrease competition between and predation upon freshly transplanted corals, and therefore optimize the long-term outcomes of the restoration process [21,72].

*4.3. Coral Restoration Influences the Composition of the Benthic Community*

Restoration affected the composition of benthic communities at all four locations, highlighting that coral restoration efforts can affect communities at a much greater scale than that of the coral transplants. This result supports the idea that characterizing restoration effectiveness requires broad reef-scale considerations [4]. Restoration methodologies, from site selection to the use of artificial structures and the species and density of coral transplants used, all require careful consideration in terms of their impact on local benthic communities. Site selection, in particular, is increasingly recognized as an important factor for maximizing the outcomes of restoration efforts [10,21,72]. Comparison of the benthic community composition between restored and control reference sites is a useful indicator of the appropriateness of the site selected. For example, similarities in benthic community assemblages between restored and control reference sites in the Florida Keys suggest that restoration efforts increased the resilience of benthic communities at these sites, and that site selection for the restoration effort was indeed appropriate, even given the degree of natural degradation at the control reference sites.

## 5. Limitations and Further Research

This study represents a "snapshot" of the responses of coral assemblages to restoration practices and our data and remarks on reef resilience should be interpreted within this context. Because of our sampling design comparisons of restoration effectiveness among the four programs are limited due to the variable restoration designs, the level of transplant maintenance, and the age of restored plots all varied among the four locations. Likewise, in three of the programs, only one type of restoration design was used (i.e., metal frames in the Maldives, midwater nurseries at both Caribbean locations), precluding meaningful comparisons of restoration effectiveness between designs. There is scope for small-scale research at a particular location on local indices of restoration effectiveness among different types of artificial structures and between artificial structures versus direct transplantation onto reef substrata at one location to complement our broad geographic comparisons. Furthermore, data were collected at the genus rather than species level so that restoration managers could easily replicate our monitoring program. However, species-level data, especially when quantifying coral juveniles in terms of success/recruitment, would provide greater insights into changes in coral diversity patterns and impacts on coral health, especially for restoration programs focused on restoring endangered coral species (e.g., *Acropora* species in the Caribbean). Overall, our research reveals that planting corals onto degraded reefs results in consistent long-term increases in hard coral cover and reef structural complexity, both of which are necessary steps in the recovery of degraded reefs, a major goal of restoration programs. The results presented here thus demonstrate that the potential for coral restoration efforts to increase coral reef resilience in the long-term is promising, but restoration practices should focus more closely on maximizing coral generic richness, as well as planting corals off the substrata or in low-predation areas to maximize coral health at restored sites. The effectiveness of coral restoration efforts also needs to account for its effects on other important functional groups, such as fishes [4], and further explore factors that contribute to enhancing the generic diversity, fecundity, and recruitment of juvenile corals at the restored sites. Considerations of socio-economic factors will also be critical in assessing the potential of coral restoration to contribute to resilience-based reef management [4,5].

**Supplementary Materials:** The following are available online at http://www.mdpi.com/1424-2818/12/4/153/s1, Table S1. Hard coral cover among treatments Posthoc with Tukeys' contrast on linear models, Table S2. Structural complexity among treatments. Posthoc with Tukeys' contrast on linear models, Table S3. Coral juveniles among treatment. Posthoc on Kruskal Wallis with Nemenyi test, Table S4. Coral juveniles among restored sites in Koh Tao. Posthoc on Kruskal Wallis with Nemenyi test, Table S5. Coral generic richness among treatments. Posthoc with Tukeys' contrast on general linear models and Kruskal Wallis with Nemenyi test, Table S6. Coral health prevalence among treatments. Posthoc with Tukeys' contrast on general linear models, Table S7. Coral disease prevalence among treatments. Posthoc with Tukeys' contrast on general linear models, Table S8. Prevalence of compromised coral colonies among treatments. Posthoc with Tukeys' contrast on general linear models, Table S9. Prevalence of predated upon coral colonies among treatments. Posthoc with Tukeys' contrast on general linear models, and Kruskal Wallis Nemenyi tests, Table S10. Pairwise ADONIS investigating the compositional differences in coral assemblages among restoration treatments at the four program locations calculated from Bray-Curtis distance matrices, Section S2. Effects of benthic attributes on the compositional differences of coral assemblages.

**Author Contributions:** Study design, M.Y.H., R.B., A.B., N.M.G., N.M., B.L.W.; Data collection, M.Y.H., T.L.B., J.L., C.M.S., L.T.; On-site logistics for site selection and data collection, J.L., T.L.B., C.M.S., L.T.; manuscript writing, M.Y.H., R.B., A.B., N.M.G., N.M., B.L.W.; manuscript review and editing, all authors. All authors have read and agreed to the published version of the manuscript.

**Funding:** This research was supported by the College of Science and Engineering at James Cook University and the ARC Centre of Excellence for Coral Reef Studies. In the Florida Keys, data were collected following requirements of the Coral Restoration Foundation's research policies (ID CRF-2016-23), and CRF's permit FKNMS-2011-159-A4.

**Acknowledgments:** We would like to thank P. Urgell, K. Magson, F. Couture, S. Stradal, K. Ripple, R. Willis, K. Lewis, K. Kopecki, and J. Blomberg for assistance in data collection, as well as T.J. Chase for useful comments and improvements to the manuscript.

**Conflicts of Interest:** The authors declare no conflict of interest.

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
