# Peer review of "Coral Restoration Effectiveness: Multiregional Snapshots of the Long-Term Responses of Coral Assemblages to Restoration"

_diversity, doi:10.3390/d12040153_

Round 1

Reviewer 1 Report

As the authors rightly point out, despite the myriad stated goals of coral-restoration programs around the world, restoration success is typically only evaluated based on number of outplanted corals and outplant survival. This study is one of the first studies to evaluate the broader results of restoration on reef health. Given the increase in coral restoration around the world in recent years, the study is both timely and an important contribution to the coral-management community. The manuscript is well-written, and my suggestions are mostly minor (outlined in the specific comments below). I just have two more substantial concerns that I would like to see addressed prior to publication.

First, I would like to see the health metrics described in a bit more detail, particularly the “other signs of compromised health”. This metric is rather vague and seems to lump everything from coral bleaching (which should definitely be its own category), to sedimentation, to algal overgrowth and anything else. Based on the Discussion, it seems like the researchers did assess these problems more specifically in the surveys, so I would suggest that they break out the categories a bit more in their analysis and description of the results. This, I think, will allow a more comprehensive discussion of the benefits/pitfalls of various restoration techniques with regards to these stressors.    

My second concern is about the recruitment metric. As the authors point out in the Discussion, only looking at corals >5 cm is missing most of the coral recruitment. Even more importantly, I question whether the recruitment at Koh Tao is really recruitment or is just reflecting outplanting of juvenile-sized colonies? Can the authors rule this possibility out? If you can’t be sure that you’re actually measuring recruitment then I would consider removing this indicator from your study or at the very least changing the language throughout to describe this metric as “abundance of juvenile corals” rather than “recruitment”.

Specific comments:

L56: see also Toth & Kuffner 2016 for a discussion of the limitations of coral cover as a reef metric: Kuffner, Ilsa B., and Lauren T. Toth. "A geological perspective on the degradation and conservation of western Atlantic coral reefs." Conservation Biology 30.4 (2016): 706-715.

L59: I would suggest citing Alvarez-Filip et al. 2009 here: Alvarez-Filip, Lorenzo, et al. "Flattening of Caribbean coral reefs: region-wide declines in architectural complexity." Proceedings of the Royal Society B: Biological Sciences 276.1669 (2009): 3019-3025.

L62-65: Can you provide some citations supporting these statements?

L72: add “coral” before “benthic cover” and “recruitment”

L87: I’d consider changing “coral assemblages” to simply “reefs” because some of the metrics (especially structural complexity and recruitment) are indicative of broader changes to the reef, not just to the living coral assemblages. It should also be changed later in the paragraph at L93.

L105: delete the comma after Koh Tao

L114/L174: It’s worth mentioning somewhere that Coral Restoration Foundation also used seafloor table nurseries in the early years of their program.

L125: Can you provide a citation that documents the land-based and marine pollution at this site?

L164: Florida’s reefs also experienced bleaching in 1987, 1997, 1998, and 2005

L170: Delete comma after A. cervicornis

L2018: Were soft corals/gorgonians also identified to the genus level? What about sponges and zoanthids?

L223: Please briefly describe the criteria used to determine the structural complexity ranking here.

L226: Again, just hard corals or also soft corals/gorgonians?

L226: I would just list all seven categories. Were all diseases lumped together or were different diseases identified?

L250: Could you provide these model comparisons in the supplementary? Were the GLMs always the models with the lowest AIC?

L252: The Kruskal-Wallis test does not allow the same flexibility in assigning fixed versus random effects in the model. How was this accounted for? Why not use a different (non-Gaussian) distribution as you did for generic richness and recruitment? Another option would be to rank the data before running the GLM which is equivalent to running a non-parametric test, but with the more robust model.

L284: I would rephrase this to something like “to the differences in the composition of coral assemblages between sites at each location”

L288: (and throughout the Results and Discussion sections). I would change “treatments” to simply “sites” here because treatment implies that you manipulated the conditions, which is not the case.

L315-318: The wording of this sentence is a bit confusing. I would simplify to something like:  Scores at restored treatments were consistently greater than the average score across all sites (2.5 out of 5), whereas scores at unrestored treatments were consistently lower (Fig.4).

L330: Is this because they outplant juvenile-sized colonies at this site or is this truly reflective of recruitment? If you can not be sure this is actual recruitment, I would at the least rename this indicator, but maybe consider removing this indicator.

L339: add “significantly” before “differ”. You should also report that the juvenile densities were actually lower at the restored sites even though this is counter to expectations.

L355: The difference in richness between the restored site and the control reference does not seem like it’s significant based on your standard errors. Are you sure this result is correct?

L356: I would also reference Fig. 7 again here.

L368-372: I don’t understand the second half of this sentence: “which was driven by…” how is this result different from the result at the beginning of the sentence? Aren’t these saying the same thing?

L372-374: If the corals weren’t diseased or suffering from predation, what was wrong with them? What does compromised mean?? I think it’s really important here to describe what exactly you observed rather than using the vague term “unhealthy”.

L420-421: rather than repeating this line in your description of each location, I would just say once at the end that this was done for all locations.

L435: delete “and”

L437: What do you mean by “rocks”? Bare carbonate substrate (likely with turf algae)?

L450: delete “level of”

L451: change “being” to “which was”

L454: Stephanocoenia intersepta, correct? There is only one genus in this family in the western Atlantic and you identified hard corals to the genus level so I would be more specific here. Similarly, I understand why you might use Acroporidae for the Pacific sites, but for the western Atlantic, these are all Acropora spp. and I would say so.

L481: “design” should be “designs”

L500-501: perhaps something like “overall reef health” rather than “coral assemblages”. Again, structural complexity and recruitment are not measures of “coral assemblages”

L505-506: The restored sites did not always more closely resemble healthy reference communities in the study and I would argue that in terms of species composition they don’t resemble historic healthy communities either, particularly for the western Atlantic sites that focus mainly on A. cervicornis restoration.

L513: I’d say “populations” rather than “assemblages” here

L525: I would delete “Consistently”

L551: Doesn’t the following sentence suggest that these sites are limited by recruitment?

L574: I’d add something like “restoration-site” before “maintenance”

L606: I’d add something like “the idea/conclusion” before “that”

L611-612: I’d note that in many places (like the Florida Keys) even the healthiest “control reference sites” are still significantly degraded compared with natural baselines.

Supplementary

L245: Need an “s” at the end of “Florida Keys”

Reviewer 2 Report

Review: Coral restoration effectiveness: multiregional snapshots of the long-term responses of coral assemblages to restoration.

General comments: This is a landmark study that for the first time examines long term response of a variety of qualitative and quantitative aspects of several major coral restoration projects around the world. The coral restoration field has long had issues with standardization of metrics for success for reef restoration efforts, and a lack of experimental design needed to answer fundamental questions. This study makes tremendous strides towards standardization of several metrics (coral cover, structural complexity, coral diversity, coral health and coral recruitment), compared across manipulated, unmanipulated and reference/control sites. This brings much needed scientific rigor to examine some of the largest restoration programs in the world.  The results are clear, with these programs having a positive effect to varying degrees.  The manuscript is extremely well written, and I have only a few suggestions for how it might be improved.  Firstly, the manuscript could use more thought on the introduction and overall conclusions.  One angle that you might try is to set up in the introduction that non-standardizable approaches and disparate restoration goals have been a major obstacle and this study makes an important first step by providing a framework to begin to compare diverse approaches…. Another line of discussion is that reef restoration has moved from a field lacking data and methods, with a lot of bold promises, but with this framework we can begin to document positive feedback loops. A discussion of scale and cost seems to be conspicuously omitted. Clearly this is hard to quantify but it should be at least referenced and given though about how to integrate into the equation. I would very much like to see more studies that use this study as an example for future work, so I think it is worth the time to add a bit more polish to the big picture set up and conclusion.  My recommendation is publication after very minor revision of the introduction, discussion, and abstract.  I also have only a few minor comments bellow.

Specific comments:

Abstract: The final sentence of the abstract needs more thought… this is perhaps the most important sentence in the paper and the major conclusion of the study. As it stands it seems like the major takeaway is different programs have different results… I would emphasize instead how this study provides a rigorous framework an applies it to diverse approaches finding qualitative and quantitative measures for efficacy, but this needs to be expanded and many effects may be longer term and larger scale than previously thought.

Introduction: A paragraph reviewing lack of standardization and lack of follow through and monitoring of reef restoration projects would be a good addition. The field has moved from snake oil salesmen and wild wild west to restoration science and this study is a major step in that direction in my opinion.

Materials and methods: Line 227: describe the health categories in the next sentence and move the sentence about generic richness to a later point…

Results: The takeaway message of the NMDS plots is basically that each site is different, but are there any commonalities or generalizations that could be made from these plots? Are these useful and if so, how do they indicate metrics for success?

Discussion: The closing of the discussion seems to run out of steam and hesitates to draw major conclusions.  There is a lot of variation between projects but the ability to be able to compare them all in a rigorous framework opens up a lot of new and exciting possibilities.  If you had to make management recommendations based on these projects, which methods show the most promise? How might future studies determine how much variation is methodological vs. environmental?  The conclusions seem to fall a bit flat and could really close with a stronger message.
